# Advances in Loading Techniques and Quality by Design for Fused Deposition Modeling in Pharmaceutical Production: A Systematic Review

**DOI:** 10.3390/ph17111496

**Published:** 2024-11-07

**Authors:** Yusra Ahmed, Azza A. K. Mahmoud, Krisztina Ludasi, Tamás Sovány

**Affiliations:** Institute of Pharmaceutical Technology and Regulatory Affairs, University of Szeged, Eötvös u 6, H-6720 Szeged, Hungary; yousrabakri@googlemail.com (Y.A.); azzaasimkhalid@gmail.com (A.A.K.M.); ludasi.krisztina@szte.hu (K.L.)

**Keywords:** 3D printing, drug loading, fused deposition modeling, fused filament fabrication, hot-melt extrusion, quality by design

## Abstract

Background/Objectives: Three-dimensional printing technology has emerging interest in pharmaceutical manufacturing, offering new opportunities for personalized medicine and customized drug delivery systems. Fused deposition modeling (FDM) is highly regarded in the pharmaceutical industry because of its cost effectiveness, easy operation, and versatility in creating pharmaceutical dosage forms. This review investigates different methods of incorporating active pharmaceutical ingredients (APIs) into filament matrices for use in fused deposition modeling (FDM) 3D printing. Methods: Two electronic databases, the Web of Science and PubMed, were utilized to survey the literature. The selected keywords for this review were as follows: fused filament fabrication OR fused deposition modeling OR FDM OR FFF AND 3D printing AND loading techniques OR impregnation techniques AND solid dosage form. Results: This paper evaluates various loading techniques such as soaking, supercritical impregnation, microwave impregnation, and hot-melt extrusion, focusing on their effectiveness and capacity for drug incorporation. Additionally, this review includes a thorough risk assessment of the extrusion process using Ishikawa and SWOT analyses. Conclusions: Overall, this review provides comprehensive insights into the latest advancements in 3D printing for pharmaceutical applications and identifies key areas for future research and development.

## 1. Introduction

Three-dimensional printing is a recent technology which is commonly used in different fields due to its many advantages that include cost minimization, customization, and versatility in the manufacturing process by using digital designs to build 3D objects through sequentially layering compounds [1].

In the pharmaceutical sector, 3D printing has emerged as a promising tool for developing personalized dosage forms, medical devices, and drug delivery systems. By leveraging the capabilities of 3D printing, researchers and pharmaceutical companies can customize drug formulations to meet individual patient needs, optimize drug release profiles, and improve treatment outcomes [2].

The earliest machines for additive manufacturing introduced by 3D systems used stereolithography (SLA) technology [3]. Since then, numerous other techniques have been developed, including selective laser sintering (SLS), laminated object manufacturing, and fused deposition modeling (FDM) [4].

An example of innovative drug delivery systems using 3D printing technology is “Spritam^®^” which was developed by Aprecia Pharmaceuticals and was approved by the FDA in 2015. This system is manufactured with Zipdose^®^ technology, which features a drop-on-powder 3D printing method [5]. Until now, this is the only approved 3D-printed dosage form worldwide, which is in accordance with their unclear regulatory status and quality requirements.

Among the various techniques, FDM is the most studied by pharmaceutical researchers because it is more cost-effective, easier to operate, and more versatile in creating pharmaceutical dosage forms compared to SLS and SLA. This makes FDM a preferred choice for efficient and economical production of versatile dosage forms, facilitating broader applications and innovations in pharmaceutical manufacturing [6]. Furthermore, FDM is a simple, usually solvent- and powder-free, printing process, which requires limited post-processing, making it suitable for applications in medical centers (hospitals and pharmacies) [7]. This process is carried out layer by layer by thermoplastic filament extrusion, which controls the feeding of continuous filament into a heated nozzle, melting, and deposition onto a building platform, to produce 3D objects, which are designed in CAD software. This process enables accurate and cost-effective production of complex shapes and functional prototypes [8]. The ability to design complex geometries with high accuracy at a micro-scale can be effectively utilized, for example in the manufacturing of microneedles, where FDM allows the customization of needle shapes, sizes, and patterns to meet specific medical and pharmaceutical applications [9].

Filament type is one of the critical factors that affects the properties and quality of 3D-printed objects. Different materials can be used such as poly(lactic acid) (PLA), polyethylene terephthalate glycol (PETG), or butadiene styrene (ABS), and each of them provides different heat resistance, flexibility, and strength for 3D-printed objects [10]. Nevertheless, independently from the applied polymer, the incorporation of drugs into filaments to ensure the optimum drug release and therapeutic efficacy is considered the most critical step in 3D printing technology [11]. Various loading techniques, such as hot-melt extrusion (HME) [12] supercritical impregnation [13] pre or post-print soaking, and microwave-assisted loading [14], have been explored to efficiently incorporate drugs into 3D-printed dosage forms, as presented in Figure 1. These techniques offer different opportunities to increase drug loading capacity, minimize degradation of sensitive compounds, and control drug release rates. Nevertheless, optimizing loading techniques for 3D-printed pharmaceuticals is essential for advancing drug delivery systems and improving patient outcomes [15]. Therefore, a comprehensive evaluation of their advantages and disadvantages was the main aim of the present review.

Defining product quality requirements in advance by applying the Quality by Design (QbD) approach is vital during the development phase as it can enable the selection of the optimal development route and the identification of formulation and process parameters, which affect the quality of pharmaceutical products [16].

In the Quality by Design (QbD) framework, Critical Quality Attributes (CQAs) and Critical Process Parameters (CPPs) are identified and managed to ensure that the final product meets the desired quality standards. Through the application of statistical tools such as Design of Experiments (DoEs), QbD enables a systematic and scientific approach to optimize formulations and process parameters, aiming to achieve the desired product attributes, eliminate batch failures, reduce deviations, and minimize expensive investigations [17].

This systematic review aims to conduct a comprehensive evaluation and risk assessment of various drug loading techniques of filaments used for FDM 3D printing.

## 2. Methods for Literature Survey and Data Extraction

Two electronic databases, the Web of Science and PubMed, were utilized to survey the literature. The selected keywords for this review were as follows: fused filament fabrication OR fused deposition modeling OR FDM OR FFF AND 3D printing AND loading techniques OR impregnation techniques AND solid dosage form.

As shown in Figure 1, the search engines delivered 231 and 69 articles from the Web of Science and PubMed, respectively. The Endnote referencing program was used to exclude duplicates from the results, which resulted in 254 articles. Finally, titles and abstracts were screened to remove irrelevant articles based on the criteria mentioned in the methods section. The excluded articles totaled 209, which resulted in 45 articles being included in this systematic review.

The inclusion criteria were as follows:

(1)Studies must utilize fused deposition modeling (FDM) as the 3D printing technology,(2)Articles must be related to the pharmaceutical sector,(3)Only original research articles will be included,(4)Publications must be in English.

Studies that do not utilize fused deposition modeling (FDM) technology and do not belong to the pharmaceutical sector were excluded. Additionally, only original research articles were considered, while other types of publications such as conference papers, review articles, etc., were also excluded. The full texts of the selected studies were screened by two independent reviewers for extraction of relevant data using a data collection sheet. The data extraction process concentrated on several key areas, including the following:

(1) Loaded drug; (2) carrier polymer; (3) loading techniques; (4) drug amount; (5) 3D printer used; (6) processing temperature; (7) release behavior; and (8) type of dosage form. The created database and the corresponding references are displayed in Appendix A.

The database was then applied to perform statistical analysis to better understand the relationships between the applied method and product quality. Nevertheless, it should be noted that several limitations hindered the proper statistical evaluation of some factors. The main limitation was the variability of how data are reported in different studies, which led to challenges in comparing results and synthesizing the findings of the present systematic review, but additionally, the data extraction process is prone to human error. Furthermore, the fact that studies with positive results are more likely to be published than those with negative or inconclusive findings also had negative effect on proper evaluation.

The statistical analysis (factorial ANOVA) was performed with Tibco Statistica v. 14.0.1. (Tibo Inc., Palo Alto, CA, USA).

## 3. Drug Loading Techniques Used in FDM Printing

Figure 2 displays the incidence of various loading techniques applied for FDM printing applications along the 45 articles included in the final review. Loading may be divided into two big groups, where extrusion-based techniques are used before, while impregnation-based techniques may be used before, after, or even during the printing process.

### 3.1. Impregnation-Based Loading Techniques

#### 3.1.1. Soaking

In pharmaceutical 3D printing, the soaking technique includes the incorporation of drugs into polymer filaments by submerging the filament in a solution containing the drug or impregnation of the 3D-printed device by the same technique. This process relies on the use of a non-solvent for the polymer filament, facilitating the absorption of the drug on filaments, which are soluble in the solvent [18].

Although the soaking method is simple and does not require heating, it has several drawbacks, which includes the use of potentially toxic solvents and low drug loading efficiency, and it is not environmentally friendly due to the significant amount of wasted drugs in the solution or dispersion. Additionally, the scaling of this method for industrial use can be quite challenging [19].

The success of this method depends on the swelling ability of the polymer when exposed to certain solvents. The duration of filament swelling significantly affects drug loading [20] and is influenced by numerous variables such as the type of organic solvent, temperature, and drug concentration [18]. Ibrahim et al. detected an enhancement in the loading % of metformin HCl when the solvent changed from absolute ethanol (0.08% loading) to a mixture of 90% ethanol with 10% water (1.40% loading) [21]. In another study, adding L-arginine and water to the soaking solution led to an increase in the solubility of glibenclamide and consequently enhanced drug loading in the filaments [22].

Soaking printed devices in a liquid suspension of nanoparticles emphasizes the significance of nanoparticle-induced drug movement in increasing drug loading. In this process, the drug passively absorbs from the nanoparticle suspension into the device. The nanoparticles were observed to stimulate drug migration towards the device, resulting in higher drug loading percentages. A two times higher drug loading was observed above the expected figure of approximately 0.1% for Eudragit RL100 (ERL) and 11–14 times higher than the expected figure of 0.005% for poly(e-caprolactone) (PCL) [20].

Cerda et al. studied the utilization of Hansen Solubility Parameters (HSPs) and HSP distances (Ra) between the drug, solvent, and filament to identify optimal combinations for high drug loading [15]. The tool focuses on achieving a high solvent–drug Ra (>10) and an intermediate solvent–filament Ra (~10). Additionally, parameters like surface roughness and stiffness were found to influence the passive diffusion of the drug into the filaments. A predictive model based on Support Vector Machine (SVM) regression demonstrated a strong correlation between Ra, filament stiffness, and the diffusion capacity of a BCS Class II drug, nifedipine (NFD), into the filaments. Through this approach, a drug loading close to 3% *w*/*w* was achieved.

The highest drug loading capacity by the soaking techniques mentioned in the reviewed publications was 5% (*w*/*w*) [18,23].

#### 3.1.2. Microwave-Assisted Impregnation

The microwave-assisted loading method in pharmaceutical 3D printing applies microwave irradiation to enhance the loading of drugs into polymeric matrices. Microwave radiation, known for its rapid and deep penetration power, can effectively interact with polar compounds, leading to improved drug loading efficiency and potential for customized drug delivery solutions. During the soaking process, blank PVA filaments were immersed into 15 mL of acetone that was oversaturated with caffeine and subjected to magnetic stirring for 2 days at room temperature. The various samples were sealed and processed under the following conditions: the solvent temperature ranged from 40 °C to 140 °C, with a heating rate of 2–5 °C/s. The power settings changed between 1 and 400 W, using a 2.45 GHz magnetron. Different absorption levels (normal, high, very high) were applied, with stirring at 600 rpm. The contact times varied from 10 s to 20 min or cycles, and cooling was accomplished with pressurized air (>60 L/min, 2.5–4 bar). Better loading and more uniform drug distribution were observed for all the microwave processes in comparison with traditional soaking (>0.95% *w*/*w* versus 0.55% *w*/*w*, respectively) [14].

#### 3.1.3. Supercritical Impregnation

Supercritical impregnation uses carbon dioxide under supercritical conditions (scCO_2_) to penetrate, plasticize, and expand polymeric matrices. This technique enables the impregnation of polymers by dissolving the active substance in scCO_2_ and then introducing it into the polymer. Upon returning to regular atmospheric conditions, the solvent turns to gas and leaves the polymeric matrix, leaving the drug trapped inside. This method results in a solvent-free impregnated device and enables the loading of the solute throughout the polymer, rather than being limited to its surface [24]. Nevertheless, the high equipment cost, the complex operation, the limited scalability and material compatibility are considered to be the main disadvantages which may limit the utilization of this process [13].

High temperature and pressure lead to increased loading; for example, a maximum loading of 9% was achieved with ketoprofen at 75 °C and 250 bar. Even higher loading can be expected and was observed using extreme conditions of 75 °C and 400 bar, but products obtained under these conditions were not suitable for use because of polymer degradation [24].

Controversially, in other studies [3,25], supercritical fluid was used under pressures of 100 and 400 bar and temperatures of 35 and 55 °C to load mango leaf extract, and the CO_2_ density was increased at higher pressures, which enhanced the affinity of the compound to the supercritical phase and led to decreased loading. Nevertheless, the CO_2_ density decreased with increasing temperature, leading to an increased amount of compounds in the supercritical phase and subsequently higher loading levels, so the highest loading of 3% was achieved at a 55 °C temperature and 100 bar pressure.

#### 3.1.4. Coordinated 3D Printing and Liquid Dispensing

Finally, Okwuosa et al. [26] modified a dual FDM 3D printer to have two FDM nozzle heads. The right extruder head was replaced with a syringe-based liquid dispenser equipped with either a 2 mL or 10 mL syringe. Two printing modes were used: single-phase printing by alternating the deposition of the core liquid and the shell filament and multi-phase printing by printing 75% of the bottom of the shell, followed by filling the core liquid and then sealing the shell.

The single phase involves frequent switching between the two printing heads after each layer, leading to disruption of the shell printing process, and this is solved by the multi-phase printing mode. In both modes, more than 85% of dipyridamole was released before 30 min. Also, this system can be used to extend drug release by using a water-insoluble permeable polymer, Eudragit RL, in the filament used to fabricate the shell [26].

### 3.2. Extrusion-Based Loading Techniques

In pharmaceutical manufacturing, single-screw extruders are widely used for various processes, including producing drug-loaded filaments for 3D printing. The process starts with feeding the raw material into the extruder hopper. The rotating screw then conveys the material along the extruder, where heat from the barrel and mechanical energy from the screw melts the material. The continuous rotation of the screw mixes and compacts the molten material to ensure uniformity. Finally, the material is extruded through a die at the end of the extruder, shaping it into the desired form. Once it exits the die, the extruded material is cooled and solidified. Finally, the extrudate is either cut or formed into its final product shape [27]. Single-screw extruders have several advantages: they are relatively simple in design and operation and are cost-effective compared to twin-screw extruders. Their use and maintenance is also easier [28]. However, a drawback is that the drug and the carrier polymer must be thoroughly mixed before being fed into the extruder to ensure a homogeneous output mixture. This pre-mixing is crucial because the single-screw extrusion process offers limited homogenization. Various methods were employed for this pre-mixing: Ayyoubi et al. used ball milling to blend the feed powder [29], while other studies used a mortar and pestle [30,31,32,33,34,35,36,37]. In another study, the mixing was carried out in a closed plastic container on a Maxiblend mixer at 25 rpm for at least 20 min [38].

In contrast, intermeshing the screws of twin-screw extruders can convey and mix the fed material at the same time. These screws rotate together within the barrel, creating shearing and kneading forces that promote melting, blending, and compounding of the materials [39]. Their use is associated with enhanced mixing capabilities, improved thermal homogeneity, and precise control over the extrusion process. Therefore, twin-screw extrusion is a suitable technique to produce drug-loaded filaments with optimal quality, properties, and uniform drug distribution, especially as it is compatible with a wide range of material properties, for example, viscosity [40]. On the other hand, twin-screw extruders are more complex and expensive and consume more energy during operation compared to single-screw extruders; they also require more maintenance and operational expertise and have limitations in terms of scalability for certain applications, especially for small-scale production [41]. Nevertheless, a novel pilot scale HME that used Affinisol™ 15LV as a filament produced sixty doses/minute at 200 mg (750 g/h and 233 m/h of filament) [42].

A considerable drawback of these techniques is that they operate at high temperatures. In most of the reviewed articles, the applied temperature was generally above 100 °C, which can result in drug degradation. An exception is found in reference [33], which uses two immediate-release polymers, Kollidon VA64 and Kollidon 12PF, to lower the FDM printing temperature and filaments with 3% ramipril content were produced via hot-melt extrusion at 70 °C and printed at 90 °C, and [43], which used extrusion temperatures of 80 °C–100 °C. Kempin et al. [44] used the solvent-casting method to prepare polymer films containing the drug before extrusion to enable a decrease in the required extrusion temperature. Polycaprolactone and poly (L-lactide) were dissolved in methylene chloride, while Eudragit RS and ethyl cellulose were dissolved in acetone. An ethanolic solution of quinine was then added to the mixture, homogenized, and poured onto a glass plate as a thin layer, and the volatile solvent was evaporated in a drying oven. The dried films were then cut into small pieces and fed to the extruder.

Nevertheless, the application of drug-loaded filaments opens up many ways to further tailor device properties such as mechanical strength or drug release rate. Tablets with higher drug load and infill density generally have slower drug release rates than tablets with a lower drug load and infill density [45,46,47,48,49]. Another study also found that the drug loading % has a significant effect on release rate, since the release of felodipine was slower in the case of tablets with a higher drug percentage [31]. Contrary to this, in the study of Macedo et al. [50], the drug load was considered a non-significant factor and only the aqueous solubility of the drug and polymer type had a significant effect. Also, the disintegration time and the subsequent dissolution time can be decreased significantly by adding mannitol as a plasticizer and a pore former to the filament [51].

Multiple other studies [38,52,53] also revealed that polymer types significantly impacted release rates, whereas the effect of infill percentage on release rate in most formulations was relatively minor. Specifically, the combination of hydroxypropyl cellulose (HPC) and Eudragit L100 resulted in slow drug release, with 80% of the drug released within 334 min [38], while combinations of Eudragit EPO and POLYOX™ N80 exhibited faster drug release compared to formulations with POLYOX™ N10 [52]. Nevertheless, this study also concluded that the dissolution rate of the tablets increased as the tablet thickness decreased.

Viidik L. et al. [36] also mentioned the effect of using different geometries on drug release from 3D-printed pharmaceutical dosages. The results revealed that honeycomb-patterned tablets released approximately 12% of the drug within 24 h, whereas cylinder-shaped tablets only released about 2% during the same timeframe. Kulkarni et al. [54] found that the orientation of layers significantly affects the release rate of printlets. They examined two orientations: a 45° print layer produced continuously and a 0° print layer produced in batches. At a 25% infill percentage, they observed distinct differences in release rates between the two methods. Printlets from the batch process exhibited a faster release compared to those from the continuous process. This difference is attributed to how layer orientation changes the surface area exposed to the release media. In the continuous process, where layers are oriented at 45°, the top and bottom surfaces of the printlets are tightly packed, limiting the penetration of the medium and its interaction with the matrix. Conversely, in the batch process with a 0° orientation, internal pores are open, allowing immediate contact with the surrounding medium upon introduction.

Also, a waffle shape design and honeycomb-shaped tablets were associated with faster drug release rates in comparison with traditional cylindrical tablets [55,56]. Furthermore, conventional core–shell tablets usually contain solid cores; for example, in the study of Alzahrani et al., amlodipine besylate was deposited in the shell while the tablet core contained conventional compressed atorvastatin calcium [57], but sometimes the core may contain solutions or hydrogel [58].

Finally, the printing temperature can also impact drug release from 3D-printed tablets. Tablets printed at different temperatures (e.g., 190 °C, 200 °C, 220 °C) showed variations in drug release profiles. Higher temperatures potentially promoted better drug-carrier miscibility and enhanced drug release [34].

There are multiple factors influencing the mechanical strength of 3D-printed tablets, and one of them is polymer particle size. Smaller particles tend to enhance the mechanical strength and toughness of the printed tablets, whereas larger particles can create structural weaknesses, potentially diminishing the overall mechanical properties of the printed dosage forms [35].

In addition to polymer particle size, another significant factor affecting the mechanical properties of 3D-printed pharmaceutical dosage forms is the filament diameter. The filament diameter can fluctuate based on several factors during the hot-melt extrusion process, including screw speed (higher speeds cause diameter variations), barrel temperature, die design, specific feed load, powder feed rate, and cooling rate [59]. However, based on the studies and general guidelines in the field, the optimal filament diameter is around 1.75 mm [38]. Also, hot-melt extruded polymers that are used as filaments in FDM usually have high brittleness, which could be solved by using a plasticizer and talc as an insoluble structuring agent that can enhance the mechanical properties of the polymer hydroxypropyl-methyl-cellulose acetate-succinate (HPMC AS) [60].

Overall, it can be stated that extrusion (single- or twin-screw extrusion) stands out as a superior loading technique among various methods due to its versatility, efficiency, and ability to produce tailored products with controlled characteristics. Compared to alternative loading techniques, extrusion provides a more uniform distribution of APIs, higher loading efficiency (80–100%), enhanced product uniformity, and improved scalability for large-scale manufacturing [61]. Additionally, extrusion facilitates the customization of formulations, the incorporation of multiple components, and the optimization of product performance, making it a preferred choice for loading drugs into polymer matrices for various pharmaceutical applications [30]. However, Quality by Design (QbD) plays a vital role in maintaining the quality and performance of the final product due to the huge number of variables involved in the extrusion process [62]. The Ishikawa diagram (fishbone diagram or cause-and-effect diagram) is one of the valuable tools created by Dr. Kaoru Ishikawa, and it is used to visualize the causes of a problem by ensuring that all possible factors are considered. Visualization is performed by organizing causes into categories, which helps in understanding the relationship between different elements [63]. The Ishikawa diagram for the extrusion process, as shown in Figure 3, is divided into four main groups: process parameters, material attributes, therapeutic goal, and product characteristics.

A SWOT analysis is a versatile and powerful tool that provides a comprehensive framework for evaluating an organization’s position. By systematically examining internal strengths and weaknesses alongside external opportunities and threats, businesses can develop strategic plans that enhance strengths, address weaknesses, capitalize on opportunities, and mitigate potential threats. This process aids in achieving long-term success and sustainability [64]. The SWOT diagram for extrusion is shown in Figure 4.

Comparing the impregnation- and extrusion-based loading methods, it can be stated that significantly (*p* < 0.0001) higher drug loadings could be achieved with extrusion-based methods, but the thermal stress during the loading step is significantly lower (*p* < 0.001) in the case of impregnation techniques. Nevertheless, this advantage can be utilized only in those cases where post-printing impregnation is applied; since there were no significant differences (*p* > 0.05) in the printing temperatures or drug release rates between the methods, these parameters were more related to the applied polymer or to the desired therapeutic goal, respectively.

## 4. Challenges and Future Perspectives

Examining the loading techniques for filaments and conducting a risk assessment for the extrusion process, it becomes apparent that the field of 3D printing pharmaceuticals holds considerable promise for future advancements. However, 3D printing technology faces several issues that need to be resolved before being implemented in commercial production. One of these challenges is the need for standardization and regulatory acceptance. The variability in 3D printing processes, materials, and product outcomes poses challenges in ensuring consistent quality, safety, and efficacy of 3D-printed pharmaceuticals. Regulatory bodies are still developing guidelines and standards specific to 3D-printed medicines, which can hinder the widespread adoption of this innovative manufacturing approach in the pharmaceutical industry [65].

The limited availability of suitable printing materials is another challenge which makes the complex task of the selection of suitable polymers and excipients to maintain drug stability and release properties more complicated. Additionally, new materials designed specifically for pharmaceutical 3D printing applications are still ongoing [66].

Furthermore, the scalability of 3D printing for mass production remains a challenge. While 3D printing offers the flexibility to create personalized dosage forms, scaling up production to meet commercial demands efficiently and cost-effectively is a significant hurdle [67].

Another considerable challenge in pharmaceutical 3D printing is the need to ensure the accuracy and precision of drug dosing in printed dosage forms. Achieving uniform drug distribution within the printed structure and maintaining the intended drug release profile throughout the entire dosage form can be technically demanding. Moreover, the integration of complex drug delivery systems, such as controlled-release mechanisms or combination therapies, into 3D-printed dosage forms presents additional challenges [68].

Additionally, the long-term stability and shelf-life of 3D-printed pharmaceuticals pose a challenge that requires thorough investigation. Understanding the impact of printing parameters, material properties, and storage conditions on the physical and chemical stability of printed dosage forms is essential for determining their shelf-life and ensuring consistent drug efficacy over time [69].

Moving forward, researchers should focus on refining loading techniques to enhance drug loading efficiency while minimizing extrusion-associated risks. Nevertheless, advancements in material science may lead to the development of novel polymers with improved compatibility with various drugs, facilitating smoother extrusion processes and ensuring the stability of active pharmaceutical ingredients. Moreover, integrating advanced process monitoring and control systems could enable real-time adjustments during extrusion, enhancing reproducibility and ensuring consistent product quality. Effective collaboration among academic institutions, industry, and regulatory agencies is crucial for tackling safety concerns, creating standardized guidelines, and accelerating the transition of 3D-printed pharmaceuticals from experimental phases to commercial production. Looking forward, the future of 3D printing in the pharmaceutical field offers immense potential for innovation, efficiency, and personalized medicine, paving the way for transformative advancements in drug delivery and patient care.

## 5. Conclusions

This systematic review examined various drug loading techniques employed in preparing filaments for FDM 3D printing, analyzing 45 publications to highlight differences in these methods. The most used technique is hot-melt extrusion, utilizing either a single or twin-screw extruder. Alternative methods, such as soaking, microwave, or supercritical impregnation, have limitations, particularly in terms of drug loading capacity. Despite its popularity, the extrusion process is complex and involves numerous variables, necessitating the use of risk assessment tools like the Ishikawa diagram and SWOT analysis to manage this complexity.

## Figures and Tables

**Figure 1 pharmaceuticals-17-01496-f001:**
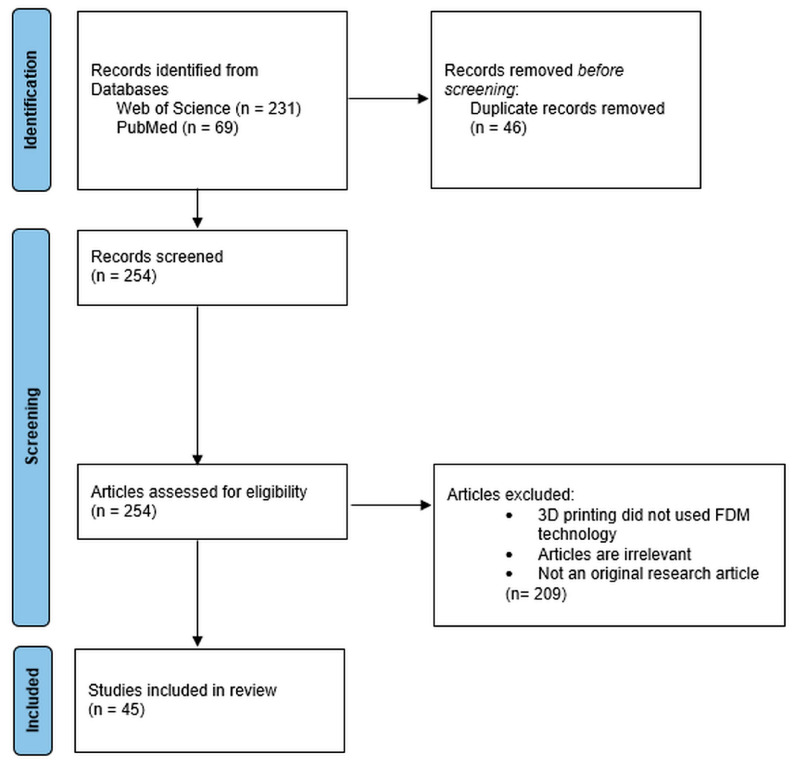
PRISMA flowchart diagram of the selection process of publications included in this systematic review.

**Figure 2 pharmaceuticals-17-01496-f002:**
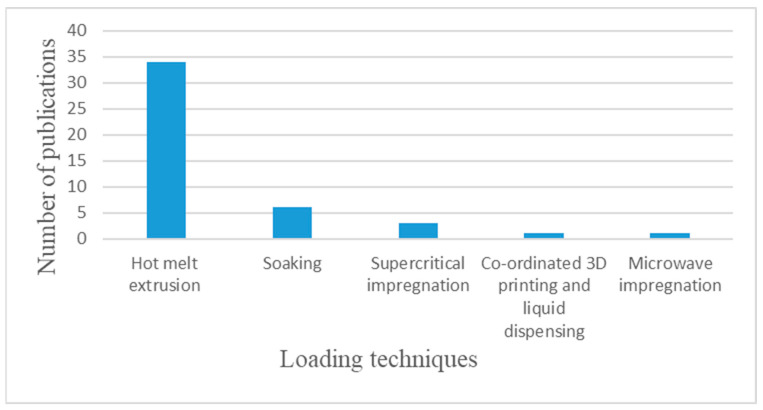
Loading techniques in reviewed articles.

**Figure 3 pharmaceuticals-17-01496-f003:**
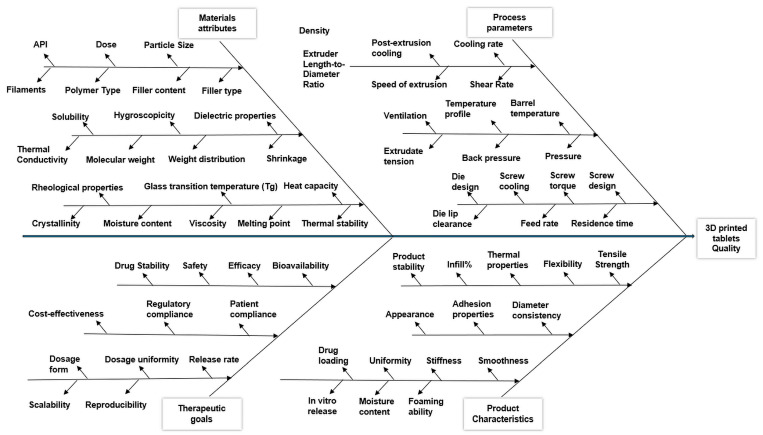
Ishikawa diagram for the extrusion process.

**Figure 4 pharmaceuticals-17-01496-f004:**
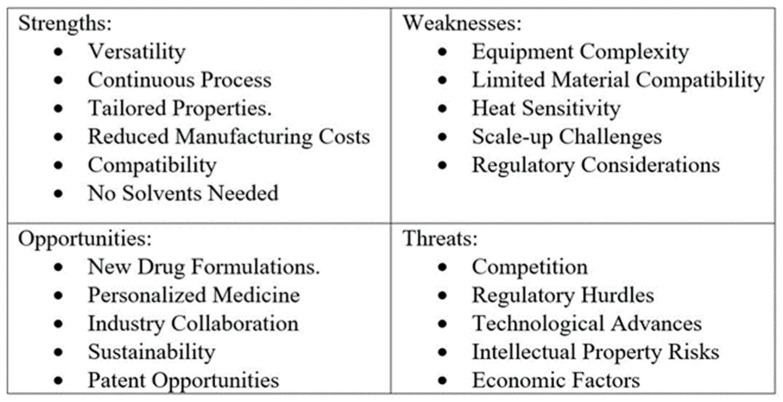
SWOT diagram for the extrusion process.

## Data Availability

The data applied in the current review paper are available in public databases.

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
