# Peer review of "Advances in Loading Techniques and Quality by Design for Fused Deposition Modeling in Pharmaceutical Production: A Systematic Review"

_pharmaceuticals, 2024, doi:10.3390/ph17111496_

Round 1
Reviewer 1 Report
Comments and Suggestions for Authors
2024-10-01
The review of pharmaceuticals-3262207, entitled ”Advances in Loading Techniques and quality by design for fused deposition modeling in Pharmaceutical Production: A systematic review”.
After careful examination of the submission, I’d like to acknowledge that it is written in acceptable English and devoted to the important problem, but it has unacceptable formatting. The systematic review has a shape of Research Article without any explained reason…
I also do not understand why references are in different brackets.
To be accepted it needs, first of all, to be reshaped in the acceptable for the Review form without inconsistencies.
Author Response
First of all, we would like to the positive opinion and valuable comments of the Reviewer which helped to improve the quality of our manuscript.
The review of pharmaceuticals-3262207, entitled ”Advances in Loading Techniques and quality by design for fused deposition modelling in Pharmaceutical Production: A systematic review”.
After careful examination of the submission, I’d like to acknowledge that it is written in acceptable English and devoted to the important problem, but it has unacceptable formatting. The systematic review has a shape of Research Article without any explained reason…
The explanation of this formatting issue was that we have used the template of the journal, which has predefined structure independently from the type of the submitted manuscript. Nevertheless, after careful checking of the style and formatting of review papers published in this journal, we have revised the formatting and changed the titles of sections to avoid misleading formatting.
I also do not understand why references are in different brackets.
We agree with the Reviewer that the formatting of references was inconsistent, which could be due to the copying of the original manuscript to the journal’s template, the comment has been considered and all citations have been revised and corrected.
Reviewer 2 Report
Comments and Suggestions for Authors
In this manuscript entitled “Advances in Loading Techniques and Quality by Design for Fused Deposition Modeling in Pharmaceutical Production: A Systematic Review,” the author's Fused Deposition Modeling (FDM) is highly regarded in the pharmaceutical industry because of its cost-effectiveness, easy operation, and versatility in creating pharmaceutical dosage forms. This review investigates different methods of incorporating active pharmaceutical ingredients (APIs) into filament matrices for use in fused deposition modeling (FDM) 3D printing. The manuscript is well-designed and written; however, several minor and major corrections are required to improve its quality.
1. Figure 1, Loading techniques in reviewed articles. Is this figure based on one-year data or overall? This should be mentioned.
2. Fused Deposition Modeling (FDM) is highly efficient for micro-needle modeling. The authors should discuss this in the introduction. Sejad et al. has done wonderful work on 3D printing of formulations https://www.sciencedirect.com/science/article/pii/S037851732100140X
3. In lines 115-116, “It appears in the enhancement of loading % of metformin HCl when the solvent changed from absolute ethanol (0.08% loading) to the mixture of ethanol 90% with water 10% (1.40% loading). What could be the reason behind it?
4. Figure 3, SWOT diagram for the extrusion process. The authors have included technological advances in the Threat section. What is the thought process behind that inclusion? As per my suggestion, it should be in the opportunity section.
5. Authors should provide a table of 3D printed products that are under clinical trial or have already been approved by regulatory agencies.
Author Response
First of all, we would like to the positive opinion and valuable comments of the Reviewer which helped to improve the quality of our manuscript.
In this manuscript entitled “Advances in Loading Techniques and Quality by Design for Fused Deposition Modeling in Pharmaceutical Production: A Systematic Review,” the author's Fused Deposition Modeling (FDM) is highly regarded in the pharmaceutical industry because of its cost-effectiveness, easy operation, and versatility in creating pharmaceutical dosage forms. This review investigates different methods of incorporating active pharmaceutical ingredients (APIs) into filament matrices for use in fused deposition modeling (FDM) 3D printing. The manuscript is well-designed and written; however, several minor and major corrections are required to improve its quality.
- Figure 1, Loading techniques in reviewed articles. Is this figure based on one-year data or overall? This should be mentioned.
This Figure (Figure 2. in the revised manuscript) represents the overall distribution loading techniques within the 45 articles included to the final revision. This was stated in lines 439-440 in the revised manuscript.
- Fused Deposition Modeling (FDM) is highly efficient for micro-needle modeling. The authors should discuss this in the introduction. Sejad et al. has done wonderful work on 3D printing of formulations https://www.sciencedirect.com/science/article/pii/S037851732100140X
We have considered the Reviewers suggestion and the role of FDM in microneedle manufacturing was added to text as follows: “The ability to design complex geometries with high accuracy at a micro-scale, can be well utilized for example in manufacturing of microneedles, where FDM allows the customization of needle shapes, sizes, and patterns to meet specific medical and pharmaceutical applications [9]”. See lines 72-76 in the revised manuscript. Regarding to the second part of the comment, the suggested work of Sejad Ayyoubi et al. was already cited as ref 29. in the manuscript.
- In lines 115-116, “It appears in the enhancement of loading % of metformin HCl when the solvent changed from absolute ethanol (0.08% loading) to the mixture of ethanol 90% with water 10% (1.40% loading). What could be the reason behind it?
The addition of water to absolute ethanol leads to an increased dielectric constant and solvent polarity, consequently increasing the solubility and hydration of hydrophilic metformin (increased hydrogen bonding between the drug and the solvent)
- Figure 3, SWOT diagram for the extrusion process. The authors have included technological advances in the Threat section. What is the thought process behind that inclusion? As per my suggestion, it should be in the opportunity section.
We agree with the Reviewer that mentioning technological advances as Threats may be strange. The rationale behind our decision is that technological advances require specialized professionals and continuous training also it relies only on digital files that threaten data security and production consistency.
- Authors should provide a table of 3D printed products that are under clinical trial or have already been approved by regulatory agencies.
As it is discussed in the manuscript, the unclear regulatory status and of lack of proper guidelines (see lines 1620-1625 of the revised manuscript) strongly hinder the commercialization of 3D printed products. Currently the Spritam tablet is the only approved 3D printed dosage form worldwide. Triastek was granted with FDA clearance to proceed for the Investigational New Drug status with several products, but the clinical trials have not started yet. Nevertheless, among drug-free medical applications such as in field of surgical implants there some approved applications.
|
Drug |
Application area |
Approval Status |
|
Drug delivery platforms (tablets) |
||
|
Spritam (Levetiracetam) |
Antiepileptic medication |
Approved |
|
T19 (Triastek) |
rheumatoid arthritis |
IND clearence |
|
T20 (Triastek) |
cardiovascular and clotting disorders |
IND clearence |
|
T21 (Triastek) |
ulcerative colitis |
IND clearence |
|
Medical device |
||
|
Osteofab |
surgical implant manufacturing |
Approved |
Reviewer 3 Report
Comments and Suggestions for Authors
Methodological Clarity: The methodology lacks detail regarding the selection criteria for included studies. While it mentions the databases used and the exclusion of non-original articles, it could be enhanced by specifying more about the process used to evaluate the quality of the studies included in the systematic review. Provide more detailed descriptions of the inclusion/exclusion criteria for the articles reviewed. Include a PRISMA flow diagram to illustrate the study selection process.
Data Extraction: Although there is some explanation of data extraction, the process seems vaguely described. More information on how data regarding loaded drugs, polymers, and other variables were handled during extraction would improve the transparency of the review.
Discussion of Limitations: The manuscript lightly touches on the limitations of the various loading techniques but does not provide an in-depth analysis of potential biases or challenges in the systematic review itself. A section discussing the review's limitations and how they were managed would enhance the paper's reliability. : Add a dedicated section that outlines the limitations of the systematic review process itself, including any potential biases in data extraction and study inclusion.
Statistical Analysis: There is limited statistical treatment of the data, despite reviewing 45 publications. Incorporating statistical comparisons between the techniques, such as effect sizes or performance metrics, would make the conclusions more robust. Consider including more statistical analysis to compare the effectiveness of different techniques. This might include quantifiable comparisons of drug-loading capacities or release profiles.
Visual Presentation: The figures, while helpful, could be enhanced for greater clarity. For example, Figure 2 (the Ishikawa diagram) and Figure 3 (the SWOT diagram) are visually cluttered and might benefit from simplification or additional explanation. Improve the figures by providing clearer labels and explanations. Consider breaking complex diagrams into multiple smaller figures to aid reader comprehension.
Comments on the Quality of English LanguageThe English could be improved to more clearly express the research.
Author Response
First of all, we would like to the positive opinion and valuable comments of the Reviewer which helped to improve the quality of our manuscript. We have performed a complete readthrough to improve the quality of the English of the paper.
Methodological Clarity: The methodology lacks detail regarding the selection criteria for included studies. While it mentions the databases used and the exclusion of non-original articles, it could be enhanced by specifying more about the process used to evaluate the quality of the studies included in the systematic review. Provide more detailed descriptions of the inclusion/exclusion criteria for the articles reviewed. Include a PRISMA flow diagram to illustrate the study selection process.
We agree with the Reviewer that the description of methodology could be improved. Detailed descriptions of the inclusion/exclusion criteria are added to the Methods section along with the PRISMA flow diagram which was replaced from the supplementary material to the main text.
Data Extraction: Although there is some explanation of data extraction, the process seems vaguely described. More information on how data regarding loaded drugs, polymers, and other variables were handled during extraction would improve the transparency of the review.
More information on how data was extracted was added to the method section as requested. See lines 428-434 of the revised manuscript.
Discussion of Limitations: The manuscript lightly touches on the limitations of the various loading techniques but does not provide an in-depth analysis of potential biases or challenges in the systematic review itself. A section discussing the review's limitations and how they were managed would enhance the paper's reliability. : Add a dedicated section that outlines the limitations of the systematic review process itself, including any potential biases in data extraction and study inclusion.
One of the limitations of systematic review is that studies with positive results are more likely to be published than those with negative or inconclusive findings; also, variability in how data is reported in different studies can lead to challenges in comparing results and synthesizing findings. Moreover, focusing only on original research articles in English may exclude valuable insights from other types of publications. In addition, the data extraction process is prone to human error. These informations were added to the revised manuscript, see lines 435-443.
Statistical Analysis: There is limited statistical treatment of the data, despite reviewing 45 publications. Incorporating statistical comparisons between the techniques, such as effect sizes or performance metrics, would make the conclusions more robust. Consider including more statistical analysis to compare the effectiveness of different techniques. This might include quantifiable comparisons of drug-loading capacities or release profiles.
Statistical analysis of the results were done as requested. The applied dataset was added to supplementary material as Table S2. It should mentioned, that the above mentioned limitations, especially the variability of data communication also hindered the proper statistical analysis.
Visual Presentation: The figures, while helpful, could be enhanced for greater clarity. For example, Figure 2 (the Ishikawa diagram) and Figure 3 (the SWOT diagram) are visually cluttered and might benefit from simplification or additional explanation. Improve the figures by providing clearer labels and explanations. Consider breaking complex diagrams into multiple smaller figures to aid reader comprehension.
We agree with the Reviewer that the visual presentation of the results could be improved. We have increased the resolution and readability of all Figures. Furthermore the Ishikawa diagram was redesigned to decrease its crowding. Nevertheless, we disagree with the Reviewers opinion that breaking of this complex diagram may aid the reader comprehensions, since the essence of this representation form is to show all important influencing parameters in whole, to express the complexity of the process.
Round 2
Reviewer 1 Report
Comments and Suggestions for Authors
It looks acceptable for me
Reviewer 2 Report
Comments and Suggestions for Authors
The authors have addressed all the queries. The manuscript can be accepted.
Reviewer 3 Report
Comments and Suggestions for Authors
The authors have addressed all comments, and I recommend publishing the article.